# *Nymphoides peltata* Root Extracts Improve Atopic Dermatitis by Regulating Skin Inflammatory and Anti-Oxidative Enzymes in 2,4-Dinitrochlorobenzene (DNCB)-Induced SKH-1 Hairless Mice

**DOI:** 10.3390/antiox12040873

**Published:** 2023-04-03

**Authors:** Tae-Young Kim, No-June Park, Hyun Jegal, Jin-Hyub Paik, Sangho Choi, Su-Nam Kim, Min Hye Yang

**Affiliations:** 1Department of Pharmacy, College of Pharmacy, Pusan National University, Busan 46241, Republic of Korea; 2Natural Products Research Institute, Korea Institute of Science and Technology, Gangneung 25451, Republic of Korea; 3International Biological Material Research Center, Korea Research Institute of Bioscience and Biotechnology, Daejeon 34141, Republic of Korea

**Keywords:** *Nymphoides peltata*, atopic dermatitis (AD), antioxidant, interleukin 4 (IL-4), immunoglobulin E (IgE), filaggrin, kallikrein related peptidase 5 (KLK5), trans epidermal water loss (TEWL), nuclear factor erythroid 2–related factor 2 (Nrf2), heme oxygenase-1 (HO-1)

## Abstract

*Nymphoides peltata* is widely used pharmacologically in Traditional Chinese Medicine and Ayurvedic medicine as a diuretic, antipyretic, or choleretic and to treat ulcers, snakebites, and edema. Previous studies have shown that phytochemicals from *N. peltata* have physiological activities such as anti-inflammatory, anti-tumor, and anti-wrinkle properties. Nevertheless, research on the anti-atopic dermatitis (AD) effect of *N. peltata* extract is limited. This study was undertaken to assess the in vitro and in vivo anti-atopic and antioxidant activities of a 95% EtOH extract of *N. peltata* roots (NPR). PI-induced RBL-2H3 cells and two typical hapten mice (oxazolone-induced BALB/c mice and 2,4-dinitrochlorobenzene (DNCB)-induced SKH-1 hairless mice) were used to investigate the effect of NPR extract on AD. The expressions of AD-related inflammatory cytokines, skin-related genes, and antioxidant enzymes were analyzed by ELISA, immunoblotting, and immunofluorescence, and skin hydration was measured using Aquaflux AF103 and SKIN-O-MAT instruments. The chemical composition of NPR extract was analyzed using an HPLC-PDA system. In this study, NPR extracts were shown to most efficiently inhibit IL-4 in PI-induced RBL-2H3 cells and AD-like skin symptoms in oxazolone-BALB/c mice compared to its whole and aerial extracts. NPR extract markedly reduced DNCB-induced increases in mast cells, epidermal thickness, IL-4 and IgE expressions, and atopic-like symptoms in SKH-1 hairless mice. In addition, NPR extract suppressed DNCB-induced changes in the expressions of skin-related genes and skin hydration and activated the Nrf2/HO-1 pathway. Three phenolic acids (chlorogenic acid, 3,5-dicaffeoylquinic acid, and 3,4-dicaffeoylquinic acid) were identified by HPLC-PDA in NPR extract. The study shows that NPR extract exhibits anti-atopic activities by inhibiting inflammatory and oxidative stress and improving skin barrier functions, and indicates that NPR extract has potential therapeutic use for the prevention and treatment of AD.

## 1. Introduction

Atopic dermatitis (AD) is a chronic relapsing inflammatory skin disease with an increasing incidence in industrialized countries. AD can start in infancy but commonly begins in adulthood [1,2]. The main symptoms of AD include pruritus, eczema, dryness, keratosis, lichenification, and erythema, which can markedly reduce the quality of life [2]. AD is mediated by various inflammatory immune cells, including macrophages, eosinophils, mast cells, dendritic cells, and T helper 2 (Th2) cells [3,4]. In particular, Th2 cells produce inflammatory cytokines such as interleukin-4 (IL-4) and interleukin-13 (IL-13) associated with high immunoglobulin E (IgE) levels and inflammatory mediator degranulation [4,5,6]. IL-4 and IL-13 cytokine overexpressions downregulate the expressions of the skin barrier associated genes filaggrin (FLG) and kallikrein (KLK) [4,7], which ultimately contribute to skin barrier damage, facilitate allergen penetration, reduce skin moisture, and exacerbate inflammation [8,9]. Therefore, current treatments for AD include emollients or humectants to moisturize skin and topical corticosteroids or calcineurin inhibitors to reduce inflammation reactions [10,11]. These agents provide effective short-term treatment, but long-term use is associated with side effects, such as itching, transient stinging, skin atrophy, acne, and burning at application sites [11,12].

Oxidative stress is defined as an imbalance between reactive oxygen species (ROS) production and antioxidant defense mechanisms and has been implicated in the pathogenesis of AD [13,14]. ROS overproduction causes lipid peroxidation, damages DNA and enzymes, and results in a loss of biological activity and damage to epidermal keratinocytes [15]. In addition, oxidative stress stimulates various inflammatory cells such as mast cells, eosinophils, and T lymphocytes, and exacerbates the symptoms of AD [14,16]. On the other hand, antioxidant enzymes such as nuclear factor erythroid-2-related factor-2 (Nrf2) and heme oxygenase-1 (HO-1) maintain redox equilibrium by regulating oxidative stress [17], and, thus, the activation of antioxidant defense enzymes provides an alternative strategy for treating AD.

*Nymphoides peltata* (Gmel.). O. Kuntze is an aquatic edible plant of the family Menyanthacea that is distributed in Central Europe and temperate and subtropical regions of Eurasia [18,19]. *Nymphoides* species, including *N. peltate,* are widely used pharmacologically in Traditional Chinese Medicine and Ayurvedic medicine [20]. According to records, it has been used therapeutically as a diuretic, antipyretic, and as a treatment for ulcers, snake bites, swelling, biliary headache, scabies, and rheumatism [20]. In addition, a paste of its fresh leaves has been used to treat chronic headaches [20]. *Nymphoides* species extract exhibits bioactivity properties such as anti-inflammatory, antioxidant, and anticonvulsant, and compounds such as flavonoids, phenolic acid, tannins, saponins, and triterpenoids have been reported in the extract [20,21,22,23,24]. Furthermore, to date, extracts of *N. peltata* have been reported to have anti-inflammatory [25], anti-tumor [19], and anti-wrinkle [26] activities, and the presence of phenolic compounds such as ephedrine, ephedradine C, 4-hydroxy-coumarin, and benzopyrone has been detected [20]. Nevertheless, no research has been conducted on the effects of *N. peltata* extract on AD. In this study, we evaluated the anti-AD and antioxidant effects of 95% EtOH *N. peltata* extracts (whole plants, aerial parts, or roots extract) on the in vitro PI-induced RBL-2H3 cells and the in vivo oxazolone-induced skin lesions in BALB/c mice and 2,4-dinitrochlorobenzene (DNCB)-induced SKH-1 hairless mice.

## 2. Materials and Methods

### 2.1. General

Nuclear magnetic resonance (NMR) spectra were determined using a JEOL 400 MHz NMR instrument (JNM-ECZ400S, JEOL, Tokyo, Japan) and DMSO-*d*_6_ was used as the solvent for NMR measurement. The chemical shifts were reported in *δ* (ppm) units and coupling constants (*J*) in Hz. Preparative high-performance liquid chromatography (prep-HPLC) was conducted using a Gilson HPLC system (GILSON Inc., Middleton, WI, USA) and Shimadzu HPLC system (Shimadzu Corporation, Kyoto, Japan) and column (Watchers 120 ODS-BP, S-10 μm, 150 × 10 mm, Isu Industry Corp., Seoul, Republic of Korea). The Gilson HPLC system was equipped with two pumps (305 master pump, 307 slave pump) and a mixer (811C dynamic mixer). The Shimadzu HPLC system was equipped with UV/vis detector (SPD-20A), pump (LC-20AT), and a system controller (CBM-20A). Silica gel 60 (70–230 mesh, Merck, Germany) was used for column chromatography and silica gel 60 F254 Art. 5715 (Merck, Germany) plates were used to perform thin-layer chromatography (TLC).

### 2.2. Plant Material and Extraction

Whole *N. peltata* and its aerial portion and roots were obtained from the Hantaek Botanical Garden foundation in Yongin-si, Gyeonggi-do, Republic of Korea, and authenticated by Dr. Jung Hwa Kang (Researcher Director). Voucher specimens (PNU-0040) were deposited in the Medicinal Herb Garden, Pusan National University. Whole plants (59 g), aerial parts (51.5 g), or roots (38.5 g) were air-dried at room temperature, ground to a powder, and then suspended in 2 L volumes of 95% ethanol (EtOH), and extracted ultrasonically twice for 90 min at room temperature. Extracts were then filtered and evaporated under reduced pressure in vacuo at 40 °C and freeze-dried to obtain *N. peltata* whole extract (NP, 3.9236 g, 6.65% extract yield), *N. peltata* aerial extract (NPA, 3.2513 g, 6.31% extract yield), and *N. peltata* root extract (NPR, 3.0271 g, 7.86% extract yield).

### 2.3. Cell Culture and Quantitative Real-Time Polymerase Chain Reaction (qPCR)

The rat basophilic leukemia cell line, RBL-2H3 cells, was purchased from the American Type Culture Collection (ATCC, Manassas, VA, USA) and seeded in Dulbecco’s Modified Eagle Medium (DMEM) containing 10% (*v/v*) fetal bovine serum (FBS), dimethyl sulfoxide (DMSO), and NP, NPA, or NPR extracts (10 μg/mL) for 30 min and then stimulated with phorbol 12-myristate 13-acetate (PMA) and ionomycin (PI) for 16 h. RNA was extracted using an RNeasy mini kit (Qiagen, Hilden, Germany) and reverse-transcribed using a RevertAid first-strand complementary DNA (cDNA) synthesis kit (Thermo Fisher Scientific, Bremen, Germany). Real-time PCR was conducted using a QuantaStudio 6 pro (Thermo Fisher Scientific) using SYBR1 Green (Power SYBR Green PCR Master Mix, Applied Biosystems, Foster City, CA, USA). The primer sets used were IL-4 (accession no. AY496861), forward 5′-ACC TTG CTG TCA CCC TGT TC-3′ and reverse 5′-TTG TGCA GCG TGG ACT CAT TC-3′; β-actin (accession no. EF156276.1), forward 5′-CAC GGC ATT GTC ACC AAC TG-3′ and reverse 5′-AAC ACA GCC TGG ATG GCT AC-3′. Transcript levels were normalized versus glyceraldehydes-3-phosphate dehydrogenase (GAPDH).

### 2.4. Animal Studies

BALB/c and SKH-1 hairless mice (6 weeks old) were purchased from Orient Bio (Seoul, Republic of Korea) and maintained under standard conditions (22 ± 3 °C and relative humidity 55 ± 5% under a 12 h light/dark cycle with food and water available ad libitum). All animal experiments procedures were reviewed beforehand and approved by the Institutional Animal Care and Use Committee of the Korea Institute of Science and Technology (KIST) (IRB code No. 2016-011, 2020-001; KIST).

### 2.5. Evaluation of Ear Swelling in the Oxazolone-Induced BALB/c Mice

The shaved inner and outer surfaces of both ears of BALB/c mice were sensitized by a single application of 20 μL of 1% oxazolone in a mixture of acetone/olive oil = 3:1 on experiment day 7. Ears were then challenged with 20 μL of 0.1% oxazolone from experiment day 8 on alternate days for 3 weeks. 1% NP, NPA, and NPR indicate that the amount of *N. peltata* extract is 1% (10 mg/mL) compared to the solution (propylene glycol/EtOH = 7:3). The mice were treated with 1% *N. peltata* extracts (NP, NPA, and NPR) twice a day for 3 weeks. Ear swelling and skin inflammation were measured on experiment day 28. On the last day of the experiment, mice were sacrificed, and ear skin was divided into two pieces for additional analysis. One piece was fixed in 3.7% formaldehyde (Sigma-Aldrich, St. Louis, MO, USA), and the other was collected by freezing at −80 °C.

### 2.6. Evaluation of AD Lesion Severity in the 2,4-Dinitrochlorobenzene (DNCB)-Induced SKH-1 Hairless Mice

The dorsal skin of SKH-1 hairless mice was sensitized with a single application of 200 μL of 1% DNCB in a mixture of acetone/olive oil = 3:1 on experiment day 7. Then, 200 μL of 0.1% DNCB with 1% NPR extract was administered from experiment day 8, 3 times/week for 2 weeks (experiment day 21). NPR extracts were treated at 4 h intervals after DNCB induction to exclude interaction between DNCB and samples. A 1% NPR extract indicates that the amount of *N. peltata* root extract is 1% (10 mg/mL) compared to the solution (propylene glycol/EtOH = 7:3). Transepidermal water loss (TEWL) and skin hydration were measured at the beginning and the end of the experiment under standard conditions using Aquaflux AF103 (Biox systems, London, UK) and SKIN-O-MAT (Cosmomed, Ruhr, Germany) instruments. On the last day of the experiment, mice were sacrificed, and skin and blood samples were collected for further analysis. The dorsal skin was divided into two pieces. One piece was fixed in 3.7% formaldehyde (Sigma-Aldrich), and the other was collected by freezing at −80 °C.

### 2.7. Histological Examination

Ear and dorsal skins of BALB/c and SKH-1 hairless mice were fixed in 10% formaldehyde (Sigma-Aldrich) for 24 h and embedded in paraffin wax. To measure changes in epidermal thicknesses and mast cell counts, tissues were serially sectioned at 4 μm and stained with hematoxylin and eosin (H&E) or toluidine blue. Epidermal thicknesses were determined using HK Basic software (KOPTIC, Seoul, Republic of Korea), and the number of mast cells infiltrating skin layers were counted in three randomly selected sections per mouse.

### 2.8. Immunohistochemistry

Paraffin tissue sections (4 μm) were deparaffinized and immunostained for kallikrein related peptidase 5 (KLK5) and filaggrin (Abcam, Cambridge, MA, USA) using Goat Anti-Rabbit IgG H&L (Alexa Fluor^®^ 488, Abcam) secondary antibodies, according to the manufacturer’s instructions. Images were acquired using a fluorescence microscope (Eclipse TE2000-U; Nikon, Tokyo, Japan).

### 2.9. Measurement of Serum IL-4 and IgE by ELISA

Blood samples were collected from the abdominal aortas, and serum was obtained by centrifuging (8000 rpm, 15 min, 4 °C), and then stored. Total serum IL-4 and IgE levels were determined using enzyme-linked immunosorbent assay (ELISA) kits (eBioscience, San Diego, CA, USA).

### 2.10. Immunoblotting Analysis

SKH-1 hairless mice dorsal tissues were lysed radioimmunoprecipitation assay (RIPA) buffer (BioPrince, Chuncheon, Republic of Korea) supplemented with protease inhibitor cocktail (Roche, Basel, Switzerland) and phosphatase inhibitor cocktail 2, 3 (Sigma-Aldrich, St. Louis, MO, USA). Total proteins (20 μg) were resolved by sodium dodecyl sulfate-polyacrylamide gel electrophoresis (SDS-PAGE) and transferred to polyvinylidene fluoride (PVDF) membranes (Immobilon, Millipore). Blots were incubated with primary antibodies, viz. IL-4 (1/1000; Abcam), KLK5 (1/1000; Abcam), phosphorylated-nuclear factor erythroid 2-related factor 2 (pNrf2) (1/500; Thermo Fisher Scientific, Waltham, MA, USA), Nrf2 (1/500; BioLegend, San Diego, CA, USA), HO-1 (1/500; Abcam), and GAPDH (1/3000; Santa Cruz Biotechnology, Santa Cruz, CA, USA), treated with horseradish-peroxidase (HRP)-linked secondary antibodies (1/5000; Santa Cruz) and visualized using an enhanced chemiluminescence (ECL) kit (Thermo Fisher Scientific). Band intensities were measured using ImageJ Ver. 1.5.2 (National Institutes of Health, Bethesda, MD, USA) and normalized versus GAPDH.

### 2.11. Isolation and HPLC Analysis of NPR Extract 

The NPR was suspended in 2 L of distilled water and then separated with hexane (Hex), chloroform (CHCl_3_), ethyl acetate (EtOAc), and *n*-butanol (BuOH) (2 × 4 L) to collect 4 liquid partitioning fractions such as Hex (NPH, 80.34 g), CHCl_3_ (NPC, 7.44 g), EtOAc (NPE, 26.61 g), and *n*-BuOH (NPB, 118.75 g). The NPE fraction (26.61 g) was applied to silica gel column chromatography using EtOAc: methanol (MeOH) (10:1 → 100% MeOH gradient) to give 7 fractions (NPE1 to NPE7). Subfraction NPE6 was applied to Gilson prep-HPLC system (UV wavelength at 250 nm; flow rate 2 mL/min) and eluted with MeOH:H_2_O (10:90 → 100% MeOH gradient) to give 4 subfractions (NPE6-1 to NPE6-4). Subfraction NPE6-1 (379.7 mg) was applied to Shimadzu prep-HPLC system (UV wavelength at 250 and 330 nm; flow rate, 2 mL/min) and eluted with a 0.1% formic acid in acetonitrile (ACN):0.1% formic acid in ultrapure water (H_2_O) isocratic system (20:80) to obtain compound **1** (3.9 mg, retention time *(t*_R_) 32 min). Subfraction NPE7 was applied to Shimadzu prep-HPLC system (UV wavelength at 250 and 330 nm; flow rate 2 mL/min) by isocratic eluted with 0.1% formic acid in ACN:0.1% formic acid in H_2_O (20:80) to give 9 subfractions (NPE7-1 to NPE7-9). Subfraction NPE7-7 (98.4 mg) was applied to Shimadzu prep-HPLC system (UV wavelength at 250 and 330 nm; flow rate 2 mL/min) and eluted with a 0.1% formic acid in ACN:0.1% formic acid in H_2_O isocratic system (22:78) to obtain compound **2** (32.3 mg, *t*_R_ 22 min). Subfraction NPE7-8 (60.1 mg) was applied to Shimadzu prep-HPLC system (UV wavelength at 250 and 330 nm; flow rate 2 mL/min) and eluted with a 0.1% formic acid in ACN:0.1% formic acid in H_2_O isocratic system (22:78) to obtain compound **3** (14.7 mg, *t*_R_ 28 min).

The high-performance liquid chromatography-photodiode array (HPLC-PDA) analysis was performed using an e2695 separation module and a 2998 PDA detector (Waters Corporation, Milford, MA, USA). Data were collected using Empower^®^ 3 Chromatography Software. NPR extract was dissolved in MeOH/H_2_O = 1:1 (*v*/*v*) at a concentration of 10 mg/mL as a sample solution. Three major compounds (chlorogenic acid, 3,5-dicaffeoylquinic acid, and 3,4-dicaffeoylquinic acid) were isolated and purified from NPR extract and identified by NMR, and MeOH/H_2_O = 1:1 (*v*/*v*) at a concentration of 1 mg/mL was used as a reference solution. Extract analysis was performed using a Sun Fire^®^ C18 column (4.6 × 150 mm, 5 μm, Waters) at 40 ± 2 °C using solvent A (0.1% formic acid in ACN) and solvent B (0.1% formic acid in H_2_O) using the following protocol: 0 to 5 min, 5 to 10% A; 5 to 10 min, 10 to 20% A; 10 to 15 min, 20 to 30% A; 15 to 20 min, 30 to 40% A; 20 to 25 min, 40 to 5% A). All injections were 10 µL in volume, and the column flow rate was set at 1.0 mL/min. The PDA acquisition wavelength range was 210 to 400 nm.

### 2.12. Statistical Analysis

The significances of differences between the untreated control and treatment groups were determined by one-way analysis of variance (ANOVA) followed by Tukey’s multiple comparisons test. Results are presented as the means ± standard deviations (SDs) or standard errors (SEs) of 2 to 3 independent experiments. Statistical significance was accepted for *p* values < 0.05.

## 3. Results

### 3.1. Inhibitory Effects of N. peltata Extracts (NP, NPA, and NPR) on IL-4 in RBL-2H3 Cells

IL-4 overexpression induced by PI treatment in RBL-2H3 cells was used to compare the anti-AD effects of the three different *N. peltata* extracts (NP, NPA, and NPR). In the result of qPCR analysis, the expression of the IL-4 mRNA gene was 0.084 ± 0.09 in the untreated control group, whereas it increased by approximately 12.5-fold in the PI treatment group (1 ± 0.36) compared to the control group. The treatment groups with NP, NPA, or NPR extracts showed a decrease in the expression of the IL-4 mRNA gene compared to the PI treatment group, with values of 0.477 ± 0.09, 0.169 ± 0.18, and 0.097 ± 0.13, respectively (Figure 1).

### 3.2. Inhibitory Effects of N. peltata Extracts (NP, NPA, and NPR) on Oxazolone-Induced Histological Changes BALB/c Mice

Oxazolone (OX) administration produced typical AD-like symptoms, which included erythema, edema, and dryness, but NPA or NPR extracts treatment improved these symptoms, and NPR extract was more effective than NPA (Figure 2A). In H&E staining, OX showed an increased mean ear (0.42 mm) and epidermal thicknesses (39.48 μm). However, the application of NPA or NPR extracts reduced ear thickness to 0.39 mm and 0.32 mm, respectively, and epidermal thicknesses to 31.32 μm and 29.86 μm, respectively (Figure 2B–D). These results showed that NPR extract was the most active extract, and thus it was used in the DNCB-SKH-1 hairless mice study.

### 3.3. Effects of NPR Extract on Histological Changes in DNCB-SKH-1 Hairless Mice

To evaluate the effect of NPR extract on AD-like skin lesions, DNCB was repeatedly applied to the dorsal skin of SKH-1 hairless mice. As shown in Figure 3A, AD-like symptoms such as skin thickening, severe erythema, edema, bleeding, excoriation, dryness, scarring, and erosion were observed in the DNCB group, but these symptoms were significantly alleviated by NPR extract application. In addition, DNCB application increased epidermal thickness and mast cell infiltration by 3.8- and 3.2-fold, respectively, versus the untreated control group. However, NPR extract treatment significantly inhibited DNCB-induced increases in epidermal thickness and mast cell infiltration by 2.2- and 1.5-fold, respectively (Figure 3B–E). 

### 3.4. Effects of NPR Extract on Serum Concentrations of IL-4 and IgE and on Skin Barrier Function in DNCB-SKH-1 Hairless Mice

We examined whether NPR extracts affected serum IL-4 and IgE concentration and TEWL and skin hydration to evaluate the anti-AD effect of NPR extract. The expression levels of serum IL-4 and IgE were increased by DNCB treatment but significantly decreased by NPR extract treatment by 34% and 58%, respectively, versus the DNCB group (Figure 4A,B). In addition, DNCB treatment increased TEWL to 74.7 g/m^2^/h versus the untreated control group (31.7 g/m^2^/h). TEWL in the NPR extract group was 55.8 g/m^2^/h, which was 25% lower than that in the DNCB group (Figure 4C). Skin moisture was 52% lower for AD mice than the untreated control group. NPR extract application improved skin moisture by 27% compared to the DNCB group (Figure 4D). 

### 3.5. Effect of NPR Extract on Skin Barrier Function Associated Proteins in DNCB-SKH-1 Hairless Mice

Immunohistochemical staining for KLK5 and FLG in the epidermis and stratum corneum showed that NPR extract treatment decreased or increased staining intensities and areas, respectively, versus DNCB-treated mice (Figure 5A,B). In DNCB-treated mice, IL-4 and KLK5 protein levels were both increased by 67% versus the untreated control group, but NPR extracts treatment significantly reduced IL-4 and KLK5 protein expression levels by 57% and 73%, respectively, versus DNCB-treated mice (Figure 5C,D).

### 3.6. Effect of NPR Extract on Antioxidant Enzyme Expressions DNCB-SKH-1 Hairless Mice

The activations of the transcription factors pNrf2, Nrf2, and HO-1 were evaluated by immunoblotting to investigate whether NPR extract activates antioxidant enzymes. DNCB treatment decreased the expressions of pNrf2 and Nrf2, but NPR extracts pretreatment significantly increased their expressions by 2.1- and 1.5-fold, respectively, versus the DNCB group (Figure 6B,C). In addition, DNCB treatment reduced the expression of HO-1 by 15%, but NPR extract pretreatment increased HO-1 expression by 38% vs. DNCB-treated mice (Figure 6D).

### 3.7. Isolation of Compounds from NPR Extract

The NPR extract was applied to silica gel column chromatography and RP HPLC to obtain three known phenolic acids. The isolated compound was identified by NMR analysis and comparison to the literature (Table 1 and Figure 7).

Compound **1** was obtained as a yellow amorphous powder. The ^1^H NMR and ^13^C spectrum of compound **1** showed the presence of aromatic ring protons at *δ* 7.00 (d, *J* = 2.1 Hz, 1H, H-2′), 6.94 (dd, *J* = 8.1, 2.0 Hz, 1H, H-6′), and 6.72 (d, *J* = 8.1 Hz, 1H, H-5′), and pair of the doublet bonds at *δ* 7.39 (d, *J* = 15.9, 1H, H-7′) and 6.14 (d, *J* = 15.9 Hz, 1H, H-8′), which indicated the presence of caffeic acid moieties. Additionally, the signals of oxymethine protons at δ 5.06 (d, *J* = 6.7 Hz, 1H, H-5) and 3.99 (d, *J* = 7.1 Hz, 1H, H-3), and methylene protons at *δ* 1.98–1.85 (m, 2H, H-2_ax_ and H-6_eq_) and 1.75–1.65 (m, 2H, H-2_eq_ and H-6_ax_) showed the presence of the quinic acid moiety. The ^13^C NMR signals of oxymethine resonances at *δ* 74.71 (C-5), 71.76 (C-4) and 71.57 (C-3), methylenes resonances at 37.88 (C-6), oxygenated carbon at δ 76.30 (C-1), and a carbonyl carbon resonance at 176.01 (C-7) are characteristic of quinic acid. The above NMR data are consistent with the literature on chlorogenic acid [27]. Consequently, compound **1** was concluded as chlorogenic acid. 

Compound **2** was obtained as a yellow amorphous powder. The ^1^H NMR and ^13^C spectrum of compound **2** showed the presence of two tri-substituted aromatic ring protons at *δ* 7.01 (dd, *J* = 5.6, 2.1 Hz, 2H, H-2′ and H-2″), 6.94 (dd, *J* = 7.2, 2.0 Hz, 2H, H-6′ and H-6″), and 6.73 (dd, *J* = 8.2, 2.1 Hz, 2H, H-5′ and H-5″), and four protons in trans-configuration for double bonds at *δ* 7.42 (dd, *J* = 15.9, 12.2 Hz, 2H, H-7′ and H-7″), 6.21 (d, *J* = 15.9 Hz, 1H, H-8′), and 6.15 (d, *J* = 15.9 Hz, 1H, H-8″), which indicated the presence of two caffeic acid moieties. Additionally, the signals of oxidized methine protons at *δ* 5.13 (dd, *J* = 7.3, 3.7 Hz, 2H, H-3 and H-5), methylene protons at *δ* 2.10 (dd, *J* = 13.7, 3.9 Hz, 1H, H-2_eq_), 2.02 (dd, *J* = 13.5, 8.0 Hz, 1H, H-6_ax_), 1.90–1.84 (m, 2H, H-2_ax_ and H-6_eq_), and a carbonyl carbon at 176.41 (C-7) showed the presence of a quinic acid moiety. The above NMR data are consistent with the literature on 3,5-dicaffeoylquinic acid [28,29]. Consequently, compound **2** was concluded as 3,5-dicaffeoylquinic acid.

Compound **3** was obtained as a yellow amorphous powder. The ^1^H and ^13^C NMR spectrum of compound **3** exhibited signals for two caffeic acids similar to the structure of compound **2**, suggesting that it is a caffeoylquinic acid derivative. However, in the quinic acid moiety, resonances for the chemical shift of 3 oxidized methine protons at *δ* 5.38 (td, *J* = 8.5, 4.4 Hz, 1H, H-3), 4.94 (dd, *J* = 8.3, 3.0 Hz, 1H, H-4) and 4.16 (s, 1H, H-5), and 4 methylene protons at *δ* 2.14 (d, *J* = 10.5 Hz, 2H, H-2_eq_ and H-6_ax_), 1.99 (d, J = 12.4 Hz, 1H, H-2_ax_) and 1.91–1.81 (m, 1H, H-6_eq_) and a carbonyl carbon at 175.42 (C-7) were observed in the ^1^H NMR and ^13^C NMR spectra of Compound **3**. The chemical shifts of the quinic acid moiety were compared with the literature data [28,29], and it was concluded that the quinic acid groups were substituted at H-3 and H-4. Thus, compound **3** was identified as 3,4-dicaffeoylquinic acid.

### 3.8. HPLC-PDA Analysis of NPR Extract

HPLC was performed using solvent A (0.1% formic acid in ACN) and solvent B (0.1% formic acid in H_2_O) using the profile 0 to 20 min, 5 to 40% A and then 20 to 25 min, 40 to 5% A at a wavelength of 325 nm. HPLC-PDA analysis, and the *t*_R_ and absorption maxima (λ_max_) of the three phenolic acids are as follows: (**1**) chlorogenic acid (*t*_R_ 9.76 min, λ_max_ 241.3 and 325.8 nm), (**2**) 3,5-Dicaffeoylquinic acid (*t*_R_ 15.41 min, λ_max_ 242.5 and 327.0 nm), and (**3**) 3,4-Dicaffeoylquinic acid (*t*_R_ 16.14 min, λ_max_ 243.7 and 327.0 nm) (Figure 8).

## 4. Discussion

Extracts of plant parts contain different metabolites and have significantly different activities [30,31]; thus, researchers in the medicinal and pharmaceutical fields tend to investigate the activities of specific parts. In recent years, plants and their extracts have emerged as alternative treatments for acute and chronic diseases, including AD [32,33]. Furthermore, numerous human clinical studies have demonstrated plant extracts and herbal products to have anti-AD effects and no noticeable side effects [33]. *N. peltata* is an aquatic perennial and a valuable medicinal herb in Ayurvedic medicine [20]. Although *N. peltata* extracts have been reported to possess significant anti-inflammatory, anti-tumor, and anti-wrinkle effects [19,25,26], no previous study has examined its anti-AD effect. Our present study showed that the extracts of whole *N. peltata* extract and its aerial and root extracts efficiently inhibited IL-4 induced in RBL-2H3 cells in vitro and attenuated AD-like skin symptoms in our oxazolone-BALB/c mice model in vivo, and that the NPR extract was the most active extract. Here, we report the inhibitory effect of NPR extract on AD-related inflammation and oxidative stress, and its beneficial effect on skin barrier function in AD-induced SKH-1 hairless mice.

AD is caused by an immune imbalance associated with increases in Th2 cells or immune cells and a Th2 response predominance [3], which results in the secretion of various allergen mediators from basophils or mast cells via IL-4 production and IgE isotype conversion [3,6]. In AD, these allergen mediators penetrate skin tissue and cause skin barrier dysfunction and epidermal changes, such as skin thickening or epidermal and dermal proliferation [6,34,35]. Furthermore, these Th2 responses and epidermal changes have been suggested as measures of pruritus severity and novel therapeutic targets in AD [36,37]. In the current study, we repeatedly applied DNCB to the dorsum skin of SKH-1 hairless mice to induce AD-like skin symptoms, such as severe erythema, lichenification, mast cell infiltration, and the overexpressions of serum IL-4 and IgE [38,39]. Notably, despite repeated DNCB application, topical NPR extract significantly inhibited the development of AD-like symptoms and downregulated the DNCB-induced overexpressions of serum IL-4 and IgE. Our results demonstrate that NPR extract can ameliorate the symptoms of AD and suppress the effects of inflammatory cytokines.

Severe disruption of the epidermal barrier function in AD reduces skin hydration, which makes it susceptible to inflammation [40]. This reduction in skin moisture can be assessed using TEWL levels, which are commonly used to evaluate skin barrier function and AD severity clinically [41]. TEWL increases are explained by KLK5 activation or an FLG deficiency or null mutation, as FLG is closely related to IL-4 and contributes to skin hydration and normalizing the pH gradient by producing a natural moisturizing factor (NMF) [42,43]; whereas, KLK5 plays an essential role in stratum corneum desquamation [44]. Thus, we confirmed the direct impact of NPR extract on epidermal barrier function in DNCB-induced mice. DNCB-induced AD reduced skin hydration and FLG expression and upregulated the expressions of TEWL, KLK5, and IL-4, but topical NPR extract significantly restored skin barrier function and epidermal protein homeostasis. Therefore, our results suggest that NPR extract ameliorates AD by regulating inflammatory cytokines and epidermal barrier homeostasis.

The Nrf2/HO-1 signaling pathway is an essential defense against oxidative stress [45]. Nrf2 prevents or protects against oxidative stress-mediated cellular damage and promotes the expressions of antioxidant-related factors, such as HO-1 [46]. Moreover, the Nrf2/HO-1 signaling pathway provides a defense mechanism that regulates chronic inflammatory responses induced by oxidative stress [47,48,49]. Previous studies suggest that the activation of the Nrf2/HO-1 pathway enhances the inflammatory response and epidermal homeostasis from oxidative stress in hapten-induced AD mice [47,50]. In our study, the topical application of NPR extract increased the expressions of the antioxidant enzymes pNrf2, Nrf2, and HO-1, which regulate DNCB-induced excessive oxidative stress. Overall, our results suggest that NPR extract ameliorates AD-related inflammation and oxidative stress by activating the Nrf2/HO-1 defense pathway.

Phenolic acids are secondary metabolites present in various foods [51]. These compounds have been reported to have various physiological activities [52,53], and chlorogenic acid and di-caffeoylquinic acid have been demonstrated to have potent free radical scavenging, anti-inflammatory, and antioxidant activities [53,54,55]. In the present study, HPLC-PDA analysis showed that chlorogenic acid, 3,5-dicaffeoylquinic acid, and 3,4-dicaffeoylquinic acid are three major components of NPR extract. Therefore, we suggest that the anti-AD effects of these three compounds be further investigated.

## 5. Conclusions

The present study demonstrates for the first time that the 95% EtOH extract of *N. peltata* (NPR) root has anti-AD effects. Of the three different parts of *N. peltata* examined (whole plant and aerial and root parts), NPR extracts most effectively suppressed in vitro PI-induced IL-4 and in vivo OX-induced AD-like symptoms. Furthermore, NPR extract application attenuated AD-like symptoms and decreased serum IL-4 and IgE levels in DNCB-induced SKH-1 hairless mice, and also improved skin barrier function and activated the Nrf2/HO-1 signaling pathway. These activities of the NPR extract were probably derived from the presence of chlorogenic acid, 3,5-dicaffeoylquinic acid, and 3,4-dicaffeoylquinic acid found by HPLC in the NPR extract. Our results suggest that NPR extract inhibits inflammatory responses and activates antioxidant enzymes, and that this extract provides new insights into the prevention and treatment of AD.

## Figures and Tables

**Figure 1 antioxidants-12-00873-f001:**
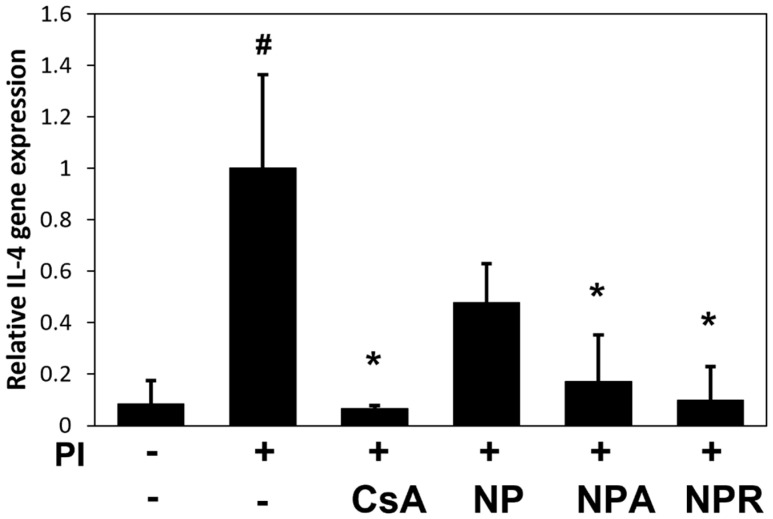
IL-4 inhibitory effect by NP, NPA, and NPR extracts. IL-4 expression was determined by qPCR versus GAPDH. Results are expressed as means ± SDs for the in vitro study of 2 to 3 independent experiments. ^#^
*p* < 0.05 vs. untreated control group; * *p* < 0.05 vs. PI-treated cells. CsA: cyclosporin-treated group; NP: PI plus *N. peltata* whole extract-treated group; NPA: PI plus *N. peltata* aerial extract-treated group; NPR: PI plus *N. peltata* root extract-treated group.

**Figure 2 antioxidants-12-00873-f002:**
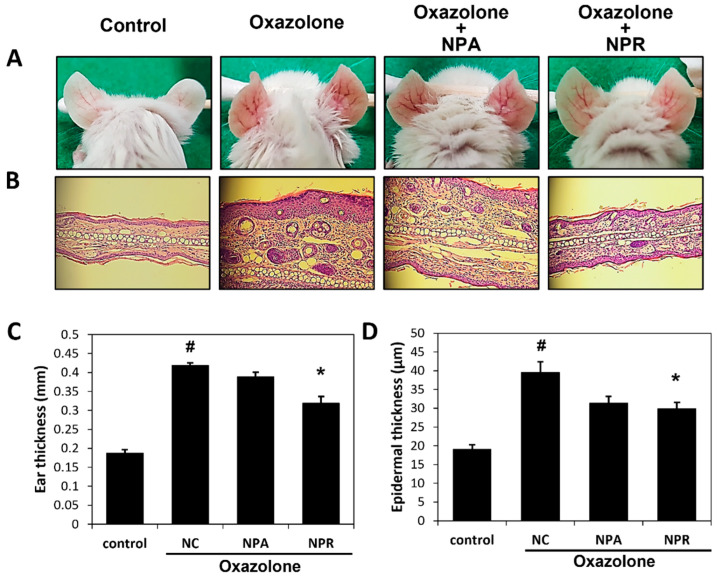
Histopathological changes by NPA, and NPR extracts. (**A**) AD clinical features of OX-induced mice, (**B**) H&E staining, (**C**) ear thicknesses, and (**D**) epidermal thicknesses of BALB/c mice after oxazolone sensitization for 4 weeks. Results are expressed as means ± SEs for the in vivo study of 2 to 3 independent experiments. ^#^
*p* < 0.05 vs. untreated control group; * *p* < 0.05 vs. oxazolone-treated mice. control: untreated control group; NC: oxazolone-induced negative control group; NP: oxazolone plus 1% *N. peltata* whole extract-treated group; NPA: oxazolone plus *N. peltata* aerial extract-treated group; NPR: oxazolone plus *N. peltata* root extract-treated group.

**Figure 3 antioxidants-12-00873-f003:**
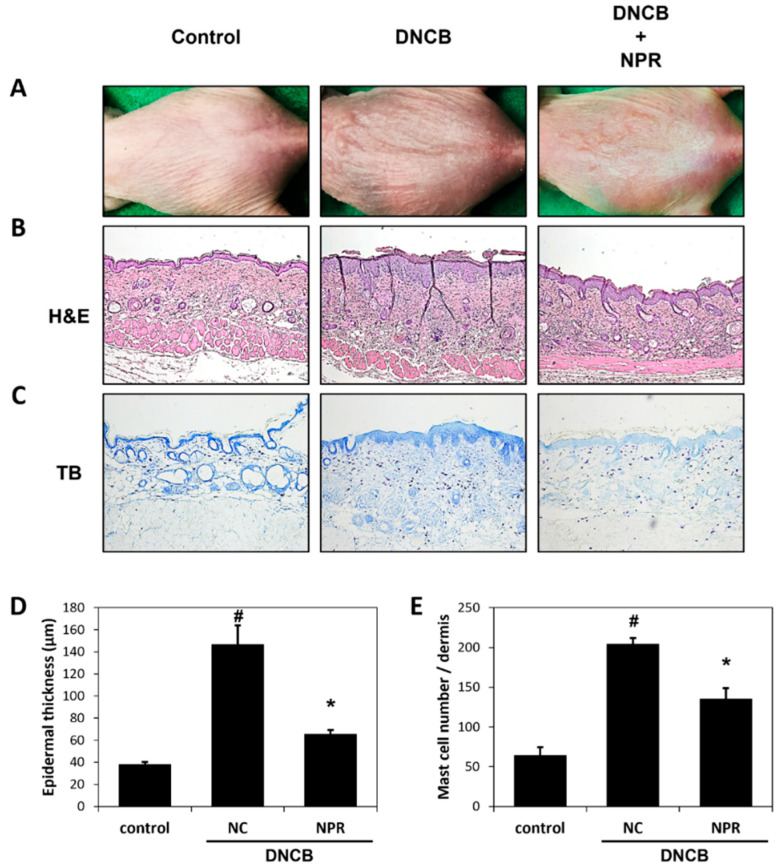
Histopathological changes by NPR extract. (**A**) AD clinical features of DNCB-induced mice, (**B**,**C**) H&E and toluidine blue staining, (**D**) epidermal thicknesses, and (**E**) mast cell numbers were measured in SKH-1 hairless mice after DNCB sensitization for 3 weeks. Results are expressed as the means ± SEs of 2 to 3 independent experiments. ^#^
*p* < 0.05 vs. untreated control group; * *p* < 0.05 vs. DNCB-treated mice. control: untreated control group; NC: DNCB-treated negative control group; NPR: DNCB plus *N. peltata* root extract-treated group.

**Figure 4 antioxidants-12-00873-f004:**
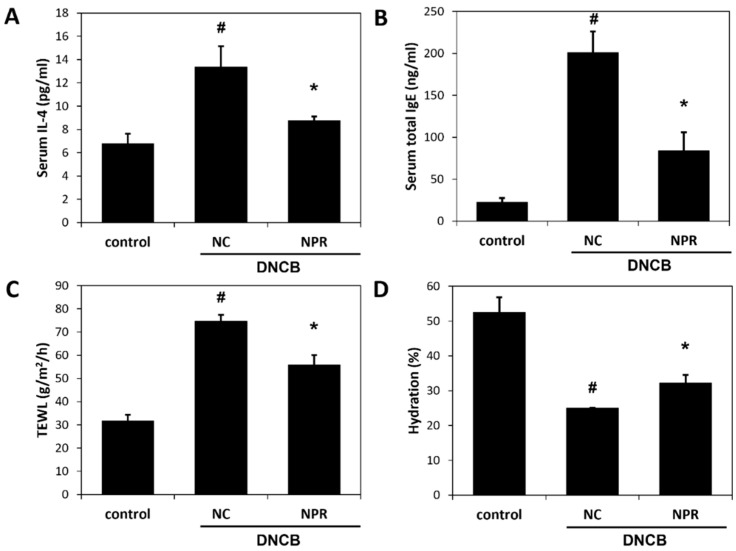
Effect of NPR extract on serum IL-4 and IgE levels and skin barrier function in SKH-1 hairless mice. (**A**,**B**) Serum total IL-4 and IgE levels were measured by collecting serum samples on the last day of the experiment using an ELISA kit. (**C**) TEWL and (**D**) skin hydration of dorsal skins were monitored during the 4-week period using Tewameter TM210 and SKIN-O-MAT instruments. Results are expressed as the means ± SEs of 2 to 3 independent experiments. ^#^
*p* < 0.05 vs. untreated control group; * *p* < 0.05 vs. DNCB-treated mice. control: untreated control group; NC: DNCB-treated negative control group; NPR: DNCB plus *N. peltata* root extract-treated group.

**Figure 5 antioxidants-12-00873-f005:**
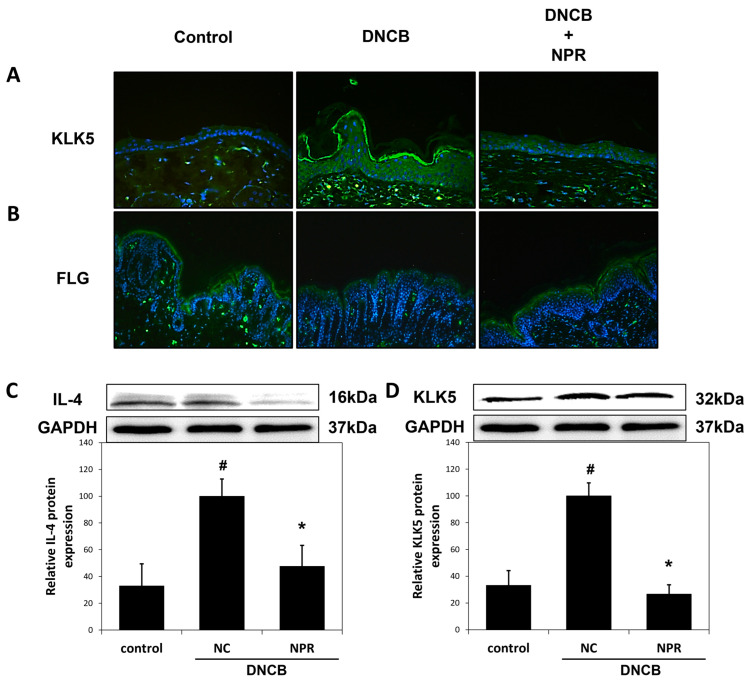
Effects of NPR extract on skin barrier function-associated proteins. (**A**,**B**) Immunohistochemical staining of FLG and KLK5, and (**C**,**D**) immunoblotting to determine IL-4 and KLK5 protein levels (normalized vs. GAPDH). Results are expressed as the means ± SE of 2 to 3 independent experiments. ^#^
*p* < 0.05 vs. untreated control group; * *p* < 0.05 vs. DNCB-treated mice. control: untreated control group; NC: DNCB-treated negative control group; NPR: DNCB plus *N. peltata* root extract-treated group.

**Figure 6 antioxidants-12-00873-f006:**
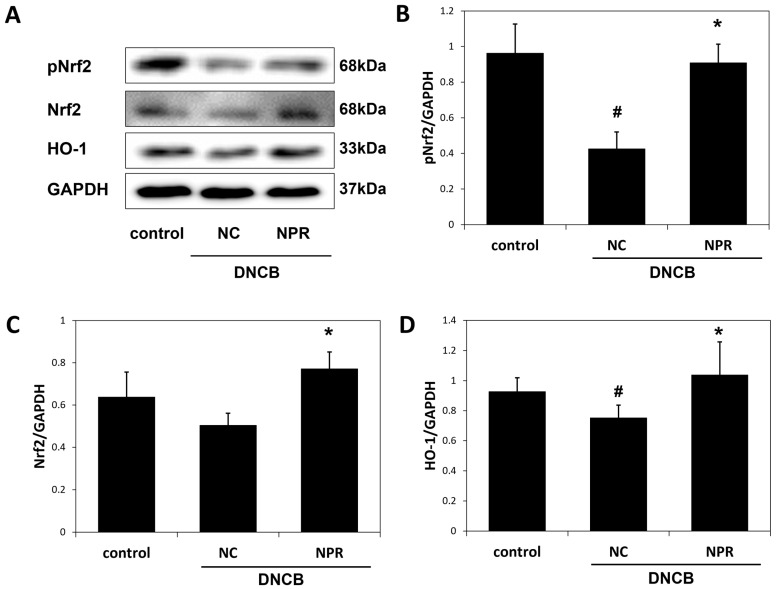
Effects of NPR extract on antioxidant defense enzyme levels. (**A**) pNrf2, Nrf2, and HO-1 protein expressions were analyzed by Western blot. (**B**) pNrf2, (**C**) Nrf2, and (**D**) HO-1 mRNA expressions were analyzed by qPCR. Results are expressed as the means ± SE of 2 to 3 independent experiments. ^#^
*p* < 0.05 vs. untreated control group; * *p* < 0.05 vs. DNCB-treated mice. control: untreated control group; NC: DNCB-treated negative control group; NPR: DNCB plus *N. peltata* root extract-treated group.

**Figure 7 antioxidants-12-00873-f007:**
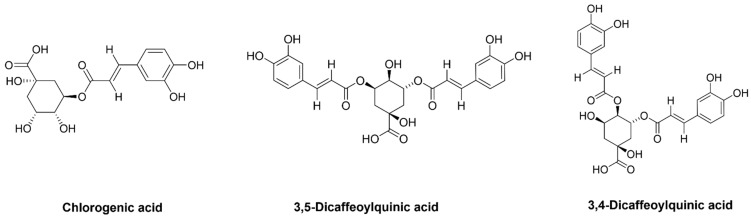
Chemical structures of the isolated compounds from NPR extract.

**Figure 8 antioxidants-12-00873-f008:**
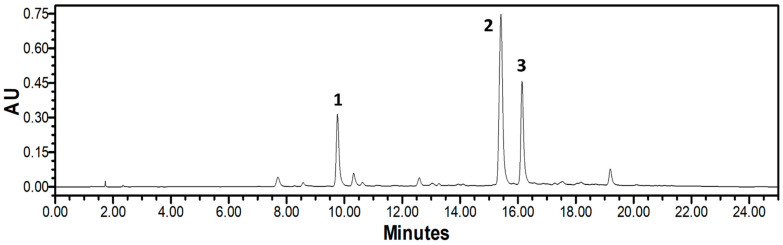
HPLC-PDA chromatogram of NPR extract at 325 nm. (**1**) Chlorogenic acid, (**2**) 3,5-Dicaffeoylquinic acid, and (**3**) 3,4-Dicaffeoylquinic acid.

**Table 1 antioxidants-12-00873-t001:** ^1^H and ^13^C NMR data (in DMSO-*d*_6_) of three major isolated compounds from NPR extract.

Position	Compound 1		Compound 2		Compound 3	
	δ_H_ (Multi, *J* in Hz)	*δ* _c_	δ_H_ (Multi, *J* in Hz)	*δ* _c_	δH (Multi, *J* in Hz)	*δ* _c_
1		76.30		73.31		74.32
2_ax_	1.98–1.85 (m)		1.90–1.84 (m)		1.99 (d, 12.4)	41.12
2_eq_	1.75–1.65 (m)		2.10 (dd, 13.7, 3.9)		2.14 (d, 10.5)	41.12
3	3.99 (d, 7.1)	71.57	5.13 (dd, 7.3, 3.7)	71.75	5.38 (td, 8.5, 4.4)	68.38
4		71.76		71.51	4.94 (dd, 8.3, 3.0)	74.32
5	5.06 (d, 6.7),	74.71	5.13 (dd, 7.3, 3.7)	71.51	4.16 (s)	68.31
6_eq_	1.98–1.85 (m)	37.88	1.90–1.84 (m)	35.54	1.91–1.81 (m)	38.10
6_ax_	1.75–1.65 (m)	37.88	2.02 (dd, 13.5, 8.0)	35.54	2.14 (d, 10.5)	38.10
7		176.01		176.41		175.42
1′		126.10		126.18		125.94
2′	7.00 (d, 2.1)	115.27	7.01 (dd, 5.6, 2.1)	116.35	7.01 (dd, 3.5, 2.1)	116.31
3′		146.11		145.61		146.04
4′		148.89		148.91		149.00
5′	6.72 (d, 8.1),	116.29	6.73 (dd, 8.2, 2.1)	115.34	6.73 (dd, 8.2, 2.1)	115.38
6′	6.94 (dd, 8.1, 2.0)	121.87	6.94 (dd, 7.2, 2.0)	122.04	6.96 (dd, 8.8, 2.1)	121.99
7′	7.39 (d, 15.9)	145.40	7.42 (dd, 15.9, 12.2),	146.11	7.45 (dd, 15.9, 12.2)	146.11
8′	6.14 (d, 15.9),	114.91	6.21 (d, 15.9)	115.04	6.21 (d, 15.9)	114.37
9′		166.48		166.82		166.57
1″				126.10		125.94
2″			7.01 (dd, 5.6, 2.1)	116.29	7.01 (dd, 3.5, 2.1),	116.25
3″				145.27		146.04
4″				148.78		149.00
5″			6.73 (dd, 8.2, 2.1)	115.17	6.73 (dd, 8.2, 2.1)	115.38
6″			6.94 (dd, 7.2, 2.0)	121.84	6.96 (dd, 8.8, 2.1)	121.90
7″			7.42 (dd, 15.9, 12.2),	146.11	7.45 (dd, 15.9, 12.2)	146.08
8″			6.15 (d, 15.9)	114.76	6.15 (d, 15.9)	114.16
9″				166.36		166.18

## Data Availability

The data are contained within the article.

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
