# Peer review of "Nymphoides peltata Root Extracts Improve Atopic Dermatitis by Regulating Skin Inflammatory and Anti-Oxidative Enzymes in 2,4-Dinitrochlorobenzene (DNCB)-Induced SKH-1 Hairless Mice"

_antioxidants, 2023, doi:10.3390/antiox12040873_

Round 1

Reviewer 1 Report

The manuscript of Tae-Young Kim et al. titled "Nymphoides peltata Root Extracts Improve Atopic Dermatitis by Regulating Skin Inflammatory and Anti-Oxidative Enzymes 3 in DNCB-Induced SKH-1 Mice" evaluates the antioxidant effects of an alcohol extract of N. peltata on oxazolone-induced skin lesions in BALB/ c and DNCB-induced SKH-1 hairless mice. Although the subject of this manuscript is interesting, the results reported in it are partial and contain many contradictions. The manuscript in this form cannot be considered suitable for publication. It would have to be heavily edited before it could be considered for possible publication.

Comments for Authors

- Authors should spell out the DNCB acronym in the title as they should at least once in the main text. 

- Authors should delete (TCM) from the Abstract and from the text. This acronym is not functional to the discussion in the text as it no longer appears after the introduction.

- Line 63: authors should double check the wording of the plant name they reported.

- There is no mention in the introduction of in vitro experimentation, which the Authors mention both in the abstract and in the main text. Please add or modify.

- The results section often refers to real-time pcr for the evaluation of the epression levels of a number of molecules. This is not reflected in the graphs presented in the figures nor in what is reported in the materials and methods. Authors should check this carefully. The pcr presented are the semi-quantitative ones. And assuming some graphs refer to real-time pcr, the sequence of primers used for those real-time pcr is not found in material and methods section.

-Section 3.1 and Figure 1:

The legend relating to Figure 1 is not at all clear. Does not match the graphical representation and text of its results. In vitro experiments are cited, but the graph and legend speak of IL4 dosage in serum.

On line 178, a real-time pcr is reported, but it appears to be represented rather semi-quantitatively. Representations of the significance are not accurate. For example, there appears to be a significance between NPA and NC and this is an important finding for downstream experiments.

Ultimately, section 3.1 should be rewritten in a linear way that overlaps with what is represented in fig.1

- Panels mentioned in the results text and figure legend are not shown in Figure 4.

Once again a qRT-PCR is indicated, but the graphs presented refer to protein expression analysis.

-The legend mentions NC subjects, but they are not shown in the figure.

- Even in Figure 5 the letters of the panel are not made explicit . Furthermore, graphs are reported which apparently refer to a pcr, but in the materials and methods no mention is made of any primers used. Instead, western blotting is reported but its densitometric analysis is missing, which the authors instead discuss in the results.

-Figure 6 shows three phenolic compounds that were isolated from NPR. The authors speculate that these may be responsible for the antioxidant effect of the treatments performed. The hypothesis is attractive, but it must be verified, because, in my opinion, it is the core of the work. The Authors should add some evaluations on the matter, even in vitro preliminarily.

- In the discussion section on line 267 the preliminary experiments of the Authors are mentioned. Which ones are they? There are no references for them.

Therefore, in consideration of what has been highlighted, I believe that the manuscript, while fundamentally potentially interesting and endowed with some originality, in this form is not yet ready to be considered for publication.

Reviewer 2 Report

The manuscript reqires major revision. Please see my comments in the attached file.

Reviewer 3 Report

The study is complex and analysed the effects against the most important AD mechanisms, signs and symptoms.

I have only one suggestion: in the future test the effects of the main compounds from the extract too.

Author Response

Review Report 3

Q1: The study is complex and analysed the effects against the most important AD mechanisms, signs and symptoms. I have only one suggestion: in the future test the effects of the main compounds from the extract too.

A: As suggested by the reviewer, research is being planned to verify the anti-atopy and antioxidant effects of the three major compounds isolated from NPR, and future research results will be reported.

Round 2

Reviewer 1 Report

The Authors have taken into consideration the suggestions provided, modifying in a satisfactory way the original version of the manuscript which, therefore, can now be considered for a publication.

Author Response

The Authors have taken into consideration the suggestions provided, modifying in a satisfactory way the original version of the manuscript which, therefore, can now be considered for a publication.

- We appreciate the reviewer's valuable comment.

Reviewer 2 Report

Some changes are required:

- Preliminary results on the toxicity of the products tested should be reported in the supplementary material.

- Lines 128-29: please ccorrect to "propylene glycol/EtOH 7:3"

- Lines 187-90: the authors did not report any detail about the isolation of these compounds. A description of the method should be added. Furthermore, no data about the identification of the same compounds by NMR are reported, and these need also to be added in the results section (spectra and description of the NMR signals).

- In their responses to my comments, the authors state that the NPR extract was administered 4 h later the skin sensitization with DNCB. However, in the Paragraph 2.5, it is reported that "200 μl of 0.1% DNCB with 1% NPR extract was administered from experiment day 8 3 times/week for 2 weeks (experiment day 21)". This needs to be clarified.

Author Response

Q1: Preliminary results on the toxicity of the products tested should be reported in the supplementary material.

A: We newly submit the supplementary material including preliminary results.

Q2: Lines 128-29: please correct to "propylene glycol/EtOH 7:3"

A: It is now changed to "propylene glycol/EtOH 7:3" in the revised version of the manuscript (line 141-142).

Q2: Lines 187-90: the authors did not report any detail about the isolation of these compounds. A description of the method should be added. Furthermore, no data about the identification of the same compounds by NMR are reported, and these need also to be added in the results section (spectra and description of the NMR signals).

A: We newly added the compounds isolation method and NMR data for the three main compounds in Section 2.11 (line 197-217), Section 3.7 (line 338-375) and Table 1 (line 376).

Q3: In their responses to my comments, the authors state that the NPR extract was administered 4 h later the skin sensitization with DNCB. However, in the Paragraph 2.5, it is reported that "200 μl of 0.1% DNCB with 1% NPR extract was administered from experiment day 8 3 times/week for 2 weeks (experiment day 21)". This needs to be clarified.

A: We now added a clear sample treatment method in Section 2.5 (line 152-153).